# The Buffering Effect of Awe on Negative Emotions in Self-Threatening Situations

**DOI:** 10.3390/bs13010044

**Published:** 2023-01-04

**Authors:** Zhaoyang Sun, Yubo Hou, Lili Song, Kun Wang, Mengchan Yuan

**Affiliations:** 1School of Psychological and Cognitive Sciences, Peking University, Beijing 100871, China; 2Beijing Key Laboratory of Behavior and Mental Health, Peking University, Beijing 100871, China; 3CAS Key Laboratory of Mental Health, Institute of Psychology, Beijing 100101, China; 4Department of Psychology, University of Chinese Academy of Sciences, Beijing 100049, China

**Keywords:** awe, negative emotions, self-threat, small-self

## Abstract

Negative emotions arising from self-threat are ubiquitous and harmful. We propose that the experience of awe awakens the small-self, which in turn alleviates these negative emotions. We examine our theoretical hypotheses in four studies employing various self-threatening situations, using distinct awe manipulations and involving participants from different countries. The participants experiencing awe reported lower levels of negative emotions arising from self-threat compared with those in the neutral (Study 1) and happiness conditions (Study 2). Moreover, we verified that the small-self mediates the alleviating effect of awe on negative emotions through measuring (Study 3) and manipulating the small-self (Study 4). Beyond a set of practical implications for promoting mental health and well-being, our research also provides novel insights into awe, self-appraisal, and self-threat.

## 1. Introduction

Individuals often encounter some experiences that create threats to their self-concept, such as failure in exams, their appearance being degraded, their status being inferior to that of former classmates, and being excluded by colleagues. Self-threat refers to the discomfort psychological state that arises when certain external information or specific situations suggest that one’s domain of the self-concept, such as power, intelligence, appearance, and social identity, is deficient [1,2]. The phenomenon occurs more prevalently and severely due to the widespread use of social media, which easily prompts people to engage in upward comparison [3,4], and the COVID-19 pandemic, which has placed people in a situation of greater disruption and frustration. Some studies have shown that self-threat can induce negative emotions, such as disappointment, sadness, depression, anger, and anxiety [1,5,6]. Most of these negative emotions have been reported in Eastern and Western cultures [5,6]. If not alleviated, these negative emotions can cause physical pain [7], aggressive behaviors [8], suicide, and other worse behaviors [9]. Therefore, answers to how to alleviate negative emotions arising from self-threat are quite important.

Strategies for alleviating negative emotions arising from self-threat involve direct resolution meant to directly compensate for the deficiencies, such as individuals who experience intelligence-threat directly enhancing intelligence [10]; fluid compensation meant to affirm self-worth in the satisfying yet threat-unrelated domain, such as individuals who experience status-threat sticking marathon-winner stickers on their cars to affirm their achievements in the field of sports [11]; distraction meant to divert attention from self-deficiency to other things, such as individuals who experience self-threat diverting attention by eating sweets [12]; and reconstructing cognition meant to change the cognition of self-deficiency, that is, altering the negative self-assessments followed by perceived self-deficiency through accepting self-deficiency or making positive attributions of self-deficiency [13,14]. Previous studies primarily focused on the first two strategies [15,16,17,18], but paid less attention to distraction and cognitive reconstruction. Notably, the direct resolution is ineffective for irreparable self-deficiency such as age and usually takes effect after a long time [17,18]. In addition, the effect of the fluid-compensation strategy is dampened if the threatened domain is the one that the individual is most proud of [2,11]. However, distraction and reconstructing cognition strategies can compensate for the above-mentioned shortcomings. We believe that there is a way, namely experiencing awe, that can echo both the need for distraction and reconstruction cognition. Awe is an emotional state that arises when individuals are confronted with something vast, immense, and beyond their current cognition of the world [19,20]. In the present research, we propose that the experience of awe is followed by a boost in the perception of the small-self, thus alleviating negative emotions arising from self-threat.

Four studies test the hypotheses that inducing awe (vs. neutral or other positive emotions) alleviates individuals’ negative emotions arising from self-threat and that the small-self plays a mediating role. Our empirical conclusions and theoretical positions promote field-specific cognition of emotions, self-appraisal, and self-threat. From the perspective of distraction and reconstructing cognition, we identify an original, malleable buffer (i.e., experiencing awe) in response to negative emotions arising from self-threat, which provides implications for coping strategies in self-threatening situations [2,17]. Furthermore, our research responds to calls for more research on awe. Prior work focused more on the effects of awe on cognition such as small-self and self-transcendence [21,22], or personal positive behavioral tendencies such as prosocial behavior [21] and global citizenship identification [23], whereas little is known about how such a unique experience contributes to mental and physical health. Our work extends previous research by pivoting to uncover that awe plays a crucial buffering role in the negative emotions arising from self-threat that shapes both mental and physical health [1,5,6].

## 2. Hypothesis Development

### 2.1. Awe and Small-Self

The vast natural landscapes, natural wonders, paranormal phenomena, and human art and artifacts can induce awe. Awe is an emotional state that arises when individuals are confronted with something so vast that it provokes a need to adjust one’s mental structure and perceived vastness vis-à-vis the self [19,20]. Keltner and Haidt proposed a prototype model explaining that awe contains two core elements, namely, the sense of vastness and the need for accommodation [19]. The former means that people realize something larger or more powerful than themselves, whereas the latter refers to the sense that people want to adjust their existing knowledge structure to adapt to new stimuli beyond their cognition, as explained by Piaget’s cognitive development theory [24].

The appraisal tendency proposes that emotion regularly arises from a certain appraisal tendency of external stimuli [25,26]. For instance, fear arises from the appraisal of high uncertainty and low control, whereas anger arises from the appraisal of high certainty and high control [27]. Some researchers have begun to identify how awe affects appraisals of the self and the social condition. Compared with neutral or other emotional states, the state of awe has been found to lead to greater comfort with adjusting mental structures to accommodate new information [20], a sense of owing more time to supporting others [28], cognition of expanding the self-boundaries, a sense of self-diminishment [19], and a greater sense of uncertainty and supernatural belief [29]. Most relevant to our research, the experience of awe contributes to the shift in self-appraisals, producing a small-self—the sense of vastness vis-à-vis the self and the sense of self-insignificance.

### 2.2. Small-Self and Negative Emotions Arising from Self-Threat

The small-self can alleviate the negative emotions arising from self-threat and be the driver of the buffering effect of awe by reducing the amount of focus on the self and changing the appraisal of the self. The small-self contains two components, namely, the sense of vastness vis-à-vis the self and the sense of self-insignificance [21,30]. The former component refers to people perceiving an entity that is larger than themselves and regarding themselves as a part of a vast entity. For instance, Shiota et al. documented that people experiencing awe are more likely to describe themselves as inhabitants on Earth instead of a certain person than those experiencing pride or happiness due to supporting greater appraisal of the small-self [20,21]. Previous research revealed that people with appraisal of the small-self tend to focus on large things rather than on themselves [30], thus being more willing to engage in collective [31], prosocial activities [21] and being more tolerant [32]. In turn, decreased self-focus promotes self-distancing, thereby leading people to ignore self-deficiencies [30,33], which alleviates some negative reactions toward self-threat [2,8]. The latter component refers to the perception that the self or one’s own daily concerns are insignificant [31,34]. The small-self makes individuals realize that they are such a small part of the world that they have little power to change the large entity no matter how perfect they are [21], thereby thinking that self-deficiencies are so insignificant that they accept them. As such, the small-self has been revealed to be effective in coping with stress [30,35]. Accordingly, individuals with an appraisal of the small-self are less likely to assess themselves negatively even if encountering self-deficiency and thus generate low levels of negative emotions [17,18]. In addition, previous research has shown that some states related to the small-self, such as self-distracting [36], self-distancing [34], and mindfulness [37], can also reduce negative reactions in self-threatening situations, which provides further support for the buffering effect of the small-self. Taken together, we propose the following two hypotheses. Figure 1 shows the conceptual model.

**H1.** 
*Awe alleviates the negative emotions arising from self-threat.*


**H2.** 
*The buffering effect of awe on negative emotions arising from self-threat is mediated by the small-self.*


### 2.3. Current Research

The current research involved four studies. Studies 1 and 2 examined the buffering effect of awe on negative emotions arising from self-threat by pitting awe against the neutral state and happiness (H1) [22,38]. Studies 3 and 4 examined H2 through measuring or manipulation of the small-self. Across four studies (see Table 1), we adopted various manipulations of emotion and different operationalization of the small-self, and implemented experiments in three types of ubiquitous self-threatening situations (i.e., appearance-threat, power-threat, and intelligence-threat) [17,18] to improve the robustness. Moreover, the groups were completely independent among these studies.

All subjects gave their informed consent for inclusion before they participated in the study. The research was conducted in accordance with the Declaration of Helsinki, and the protocol was approved by the Ethics Committee of Peking University (Project identification code #2022-02-15). All relevant data are within the manuscript and its Appendix A.

## 3. Study 1

In Study 1, we aimed to provide initial evidence for H1. This experiment was implemented in the intelligence-threatening situation.

### 3.1. Participants and Procedure

The calculated result of G*power 3.1 showed that sample size analysis of one-way analyses of variance (ANOVA) with two-level (effect size = 0.25, α = 0.05, 80% power) is at least 128 participants [39]. One hundred and sixty-four Chinese participants completed a laboratory experiment, and 13 participants were excluded because of failed attention checks. The remaining 151 participants (54 females; *M*_age_ = 27.13, *SD*_age_ = 6.28) were randomly assigned to the experimental (i.e., awe; n = 78) or the control (i.e., neutral; n = 73) condition.

After reading the informed consent, the participants assessed their baseline mood (happy, excited, lighthearted, sad, distressed, and downbeat) [40] and awe trait (“I usually feel awe,” “I see beauty all around me,” “I feel wonder almost every day,” “I often look for patterns in the objects around me,” “I have many opportunities to see the beauty of nature,” “I seek out experiences that challenge my understanding of the world”; α = 0.76) [34] on seven-point scales (1 = not at all, 7 = extremely). Second, we induced the intelligence-threat by asking the participants to answer five difficult Raven intelligence questions within 75 s (the pretest showed that less than 30% of participants answered four or five questions correctly), which was used in previous research [17,18]. Ten seconds after submitting their answers, participants received corresponding feedback: “Unfortunately, your intelligence level ranks in the bottom 10% among all the people participating in the test.” Meanwhile, we confirmed that the materials of manipulating self-threats applied in this and all the other studies could successfully induce self-threat through the pre-study (see Appendix A). Third, we induced the emotional state with a video-based task introduced by Piff et al. [21]. We presented participants with a three-minute video and then asked them to imagine themselves in the scenes and write their perceptions in five sentences. The video of the awe condition illustrated the snowy mountains, deserts, grasslands, and sweeping shots of scenic vistas in Xinjiang, China, whereas the video of the control condition illustrated a scene in which a middle-aged man with no obvious facial emotion is painting on a wooden board. (All video links are in the Appendix A). Meanwhile, at least three minutes of watching time was ensured via system supervision. Fourth, the participants rated their negative emotions (sad, distressed, and downbeat) [40] arising from self-threat (e.g., “Do you feel sad because of your intelligence based on how you are feeling right now?”; 1 = not at all, 7 = very much). We used the average of the three negative emotions for analysis [41]. Fifth, the participants completed a manipulation check by reporting their feelings (sad, fear, quiet, awe, pride, happy, and excited) on a seven-point scale (e.g., “Do you feel fear after watching that video?”) [20]. Finally, the participants reported their gender, age, and educational background.

### 3.2. Results and Discussion

#### 3.2.1. Manipulation Check

We conducted a one-way variance analysis (ANOVA) to analyze the emotions measured in the step of the manipulation check. The results showed that participants in the experimental condition reported higher awe (*M* = 5.88, *SD* = 1.37) than those in the control condition (*M* = 3.04, *SD* = 1.36; *F* (1, 146) = 155.69, *p* < 0.01, *η*^2^ = 0.52). Meanwhile, there were a few differences in pride (*F* (1, 146) = 4.39, *p* = 0.04, *η*^2^ = 0.03), fear (*F* (1, 146) = 4.14, *p* = 0.04, *η*^2^ = 0.03), and quiet (*F* (1, 146) = 6.13, *p* = 0.01, *η*^2^ = 0.04) between the two conditions, which were controlled in the following analysis [22]. No differences in the other emotions between the two conditions were found.

#### 3.2.2. Awe and Negative Emotions Arising from Self-Threat

We conducted the one-way ANOVA after controlling for the emotions (i.e., pride, fear, and quiet). The results showed that participants in the experimental condition reported lower negative emotions arising from intelligence-threat (*M* = 2.02, *SD* = 1.27) than those in the control condition (*M* = 3.46, *SD* = 1.57; *F* (1, 146) = 26.02, *p* < 0.01, *η*^2^ = 0.15).

Moreover, to provide more stable evidence, we conducted the one-way ANOVA after controlling for the emotions (i.e., pride, fear, and quiet) that differed between the two groups, baseline moods, age, gender, and educational background. The results also showed that participants in the experimental condition reported lower negative emotions arising from intelligence-threat (*M* = 2.02, *SD* = 1.27) than those in the control condition (*M* = 3.46, *SD* = 1.57; *F* (1, 137) = 26.28, *p* < 0.01, *η*^2^ = 0.16). Thus, a state of awe can alleviate negative emotions arising from self-threat, in support of H1.

#### 3.2.3. Exclude the Interference of the Trait of Awe

To examine that whether the trait of awe affects the level of the state of awe elicited by videos, we performed a moderated analysis with 5,000 iterations (Process 3.0, Model 1; Hayes, 2017) [42], in which the condition (experimental = 1, control = 0) was the independent variable, the state of awe elicited by the video was the dependent variable, the trait of awe was the moderator, and demographics were the control variables. The interaction effect of the condition and trait awe was not significant (*β* = −0.01, *t* = −0.10, *p* = 0.92), which indicated that the state of awe elicited by the video was not affected by the trait of awe.

Additionally, the results of the linear regression showed that the trait of awe did not affect negative emotions arising from self-threat (*β* = 0.04, *t* = 0.44, *p* = 0.66).

Finally, after controlling for the trait of awe, the results of ANOVA still showed that participants experiencing awe reported lower negative emotions arising from intelligence-threat (*M* = 2.02, *SD* = 1.27) than those in the control condition (*M* = 3.46, *SD* = 1.57; *F* (1, 143) = 26.81, *p* < 0.01, *η*^2^ = 0.17), supporting H1.

Study 1 suggested that awe can alleviate negative emotions arising from self-threat. This finding is in line with previous results suggesting that awe is a unique positive emotion in terms of reducing daily stress [30] and increasing one’s meaning in life [39]. Moreover, the conclusions also indicated that the trait of awe exerted no significant interference. Thus, people who are not high in the trait of awe can still have awe induced within a short period and enjoy its positive effects. This is in line with a previous study that found that the trait of awe is not associated with negative emotions arising from waiting [43]. However, Study 1 did not indicate whether awe impacts uniquely in alleviating negative emotions arising from self-threat, which was complemented in Study 2.

## 4. Study 2

In Study 2, we retested H1 by comparing awe with happiness [22] in the appearance-threatening situation.

### 4.1. Participants and Procedure

One hundred and eighty-six participants on Mechanical Turk completed the experiment. Six participants were excluded due to failed attention checks, and the remaining 180 participants (87 females; *M*_age_ = 27.60, *SD*_age_ = 6.52) were randomly assigned to the awe (n = 60), neutral (n = 59), or happiness (n = 61) condition.

First, we manipulated the appearance-threat by asking participants to think about their shortcomings of appearance and then write them down using 20–30 words, which were introduced by the studies designed by Park and Maner [44] and Choi et al. [45]. Second, we induced the emotional state with a video-based task. The participants in the awe condition watched a three-minute video that illustrated a series of expansive natural scenes, including auroras, starry skies, mountains, and the Norwegian Sea. The participants in the neutral condition watched a three-minute video illustrating an ordinary street scene in Guangdong. The participants in the happiness group watched a video illustrating a scene in which some little animals were dancing happily. The requirements during and after watching the video were the same as those in Study 1. Third, the participants rated their negative emotions arising from self-threat at the moment (as in Study 1). Finally, participants completed the manipulation check and reported their demographic information (as in Study 1).

### 4.2. Results

#### 4.2.1. Manipulation Check

The participants in the awe condition felt more awe (*M* = 5.82, *SD* = 1.26) than those in the neutral condition (*M* = 2.69, *SD* = 1.55; *F* (1, 114) = 144.48, *p* < 0.01, *η*^2^ = 0.56) and in the happiness condition (*M* = 2.49, *SD* = 1.60; *F* (1, 116) = 139.56, *p* < 0.01, *η*^2^ = 0.55). No difference in awe was found between the happiness and neutral conditions (*F* (1, 115) = 0.78, *p* = 0.38, *η*^2^ = 0.01). The participants felt happier in the happiness condition (*M* = 6.18, *SD* = 1.27) than those in the awe condition (*M* = 4.42, *SD* = 1.53; *F* (1, 116) = 52.47, *p* < 0.01, *η*^2^ = 0.31) and neutral condition (*M* = 4.32, *SD* = 1.72; *F* (1, 115) = 28.69, *p* < 0.01, *η*^2^ = 0.20). No significant difference in happiness was found between the awe and neutral conditions (*F* (1, 114) = 0.28, *p* = 0.60, *η*^2^ = 0.002). However, there was a significant difference in excitement among the three conditions (*F* (2, 175) = 18.62, *p* < 0.01, *η*^2^ = 0.18), which was controlled in the following analysis. There were no obvious differences in other emotions among the three conditions.

#### 4.2.2. Awe and Negative Emotions Arising from Self-Threat

The results of ANOVA showed that there was a significant difference in negative emotions arising from self-threat among the three groups (*F* (2, 174) = 10.93, *p* < 0.01, *η*^2^ = 0.11) (see Figure 2). Specifically, the participants in the awe condition (*M* = 3.08, *SD* = 1.19) reported lower negative emotions than those in the neutral condition (*M* = 3.90, *SD* = 1.29; *F* (1, 113) = 9.79, *p* < 0.01, *η*^2^ = 0.08) and in the happiness condition (*M* = 3.96, *SD* = 1.35; *F* (1, 115) = 14.69, *p* < 0.01, *η*^2^ = 0.11). No significant difference was found between the neutral and happiness conditions (*F* (1, 114) = 3.38, *p* = 0.07, *η*^2^ = 0.03).

Therefore, compared not only with the neutral emotion but also happiness, awe is more effective in alleviating negative emotions arising from self-threat, which indicates that awe exerts unique effects on negative emotions arising from self-threat. Thus, H1 was repeatedly supported. Next, we examined why awe could alleviate the negative emotions arising from self-threat.

## 5. Study 3

In Study 3, we aimed to examine the mediating effect of the small-self between awe and negative emotions arising from self-threat (H2). This study was implemented in an appearance-threatening situation.

### 5.1. Participants and Procedure

One hundred and sixty-two participants from Peking University completed a laboratory experiment. Fifteen participants were excluded due to failed attention checks, and 147 participants (70 females; *M*_age_ = 28.52, *SD*_age_ = 7.00) were randomly assigned to the experimental (i.e., awe; n = 81) or control (i.e., neutral; n = 66) condition.

First, we induced the appearance-threat with the same task as in Study 2. Second, we manipulated awe with a recalling task [22]. We presented participants with the same video as that in Study 1 and asked them to watch the video carefully. Then, we asked participants in the experimental condition to recall an experience during which they perceived vastness in physical or social size and felt a demand to adjust their mental structures to accommodate the new experience, and to write the experience and accompanying emotions with five sentences. Moreover, we further prompted participants that “your experience could be related, although not necessarily, to the scenes shown in the videos.” Meanwhile, we asked participants in the neutral condition to recall an experience of cleaning desks and to describe the experience and accompanying emotions in five sentences. Third, participants rated their negative emotions arising from the appearance-threat (as in Study 2). Fourth, participants were asked to fill out an eight-item and seven-point scale measuring small-self based on how they were feeling after watching the video (“I feel the presence of something greater than myself,” “I feel part of some greater entity,” “I feel like I am in the presence of something grand,” “I feel the existence of things more powerful than myself,” “I feel like my own day-to-day concerns are relatively trivial,” “In the grand scheme of things, I feel that my issues and concerns do not matter as much,” “I feel insignificant in the grand scheme of things,” “I feel small relative to something more powerful than myself”; α = 0.98; Piff et al., 2015) [21]. Fifth, participants completed the manipulation check by rating the extent to which they felt sad, fear, quiet, awe, proud, happy, and excited (e.g., “Do you feel fear after recalling that experience?”; 1 = not at all, 7 = extremely). Finally, participants reported their demographic information (as in Study 1).

### 5.2. Results and Discussion

#### 5.2.1. Manipulation Check

The participants reported higher awe in the awe condition (*M* = 5.55, *SD* = 1.52) versus the control condition (*M* = 2.42, *SD* = 1.47; *F* (1,142) = 150.53, *p* < 0.01, *η*^2^ = 0.52). The two conditions had no significant difference in other emotions.

#### 5.2.2. Awe and Negative Emotions Arising from Self-Threat

The ANOVA results showed that the participants in the awe condition reported lower negative emotions (*M* = 2.71, *SD* = 1.32) than those in the control condition (*M* = 3.67, *SD* = 1.65; *F* (1, 142) = 14.35, *p* < 0.01, *η*^2^ = 0.09), supporting H1.

#### 5.2.3. Small-Self Serves as a Mediator

We further conducted bootstrapping mediational analysis (Model 4; 5000 iterations), in which the condition (awe = 1, neutral = 0) was the independent variable, negative emotions arising from self-threat was the dependent variable, small-self was the mediating variable, and demographics were controlled. As shown in Figure 3, awe positively influenced the small-self (*b* = 0.58, *se* = 0.07, *t* = 8.59, *p* < 0.01), and the small-self negatively influenced negative emotions arising from self-threat (*b* = −0.33, *se* = 0.10, *t* = −3.42, *p* < 0.01). The small-self mediated the impact of awe on negative emotions arising from self-threat (*b* = −0.19, 95% CI [−0.35, −0.06]). The proportion of the mediating effect in the total effect was 63.74%. Meanwhile, the direct effect of awe on negative emotions arising from self-threat was not significant (*b* = −0.11, *se* = 0.10, *t* = −1.18, *p* = 0.24). Thus, the small-self played a complete mediating role. These results supported H2.

Additionally, we conducted a follow-up study in which we assessed participants’ trait of awe and baseline mood, induced the intelligence-threat, used the videos that were the same as those in Study 2, and recruited participants through Credamo. A total of 163 participants was used for empirical analysis. After controlling for the trait of awe and baseline mood, the mediating effect of small-self was also significant (*b* = −0.20, 95% CI [−0.36, −0.06]) and the proportion of it was 51.74%. Meanwhile, the direct effect of awe on negative emotions was marginally significant (*b* = −0.18, *se* = 0.10, *t* = −1.94, *p* = 0.05), so the mediating effect was almost complete.

Consistent with H2, awe stimulated the appraisal of small-self and thus alleviated negative emotions arising from self-threat, which still worked even when controlling for the trait of awe and baseline moods. The finding that a sense of self as small and insignificant ensues when people are struck with awe has received little support [20,21]; we extended this literature by examining the relevance of awe for self-threat and the role of the small-self in driving the link. Notably, since the small-self is the psychological mechanism, will the buffering effect of awe on negative emotions be inhibited if the small-self is dampened? Responses to this question would provide further support for H2, which was explored in the next study.

## 6. Study 4

In Study 4, we retested H2 by manipulating the small-self and conducted the experiment in the power-threatening situation.

### 6.1. Participants and Procedure

We implemented a 2 (emotional state: awe vs. neutral) * 2 (small-self: dampened vs. not dampened) within-subjects study. The calculation result of G*power 3.1 showed that the sample size analysis of two-way ANOVA (effect size = 0.25, α = 0.05, 80% power) is at least 196 participants [39]. A total of 270 participants recruited from Mturk completed the experiment. Eighteen participants were excluded due to failed attention checks, and 252 participants (124 females; *M*_age_ = 27.74, *SD*_age_ = 5.53) were randomly assigned to the four conditions.

After reading the informed consent, the participants received the manipulation of the power-threat via a recalling task introduced by Galinsky et al.’s research [46]. We presented participants with some words (authority, control, influence, management, manipulation, order, ruling, governance, leadership, arbitration, and control) and asked them to recall and write a circumstance in which they were oppressed by the power of others. Second, we manipulated awe through a recalling task. The video of the awe condition was the same as that in Study 1, whereas the video of the neutral condition was the same as that in Study 2. Other steps and requirements were the same as those in Study 3. Third, we manipulated the sense of small-self by asking participants to fill out the scale of small-self (as in Study 3). In order to manipulate the dampening of the small-self, the scale varied between groups. In the not-dampened group, the scale ranged from 1 to 5, whereas in the dampened group, the scale ranged from 10 to 50. Previous research indicated that answering toward the bottom of a scale would lead participants to make corresponding inferences about themselves [47]. Consequently, answering on the 10–50-point scale would cause participants to answer toward the bottom of the scale, thus dampening the small-self, compared with answering on the 1–5-point scale. This method was applied in Kim et al.’s research [48] and proved to be effective. Fourth, the participants rated their negative emotions arising from self-threat based on how they were feeling at that time (as in Study 1). Finally, participants completed the manipulation check and reported their demographic information (as in Study 3).

### 6.2. Results and Discussion

#### 6.2.1. Manipulation Check

The participants felt more awe in the awe condition (*M* = 5.30, *SD* = 1.87) versus the neutral condition (*M* = 3.53, *SD* = 1.93; *F* (1, 247) = 54.42, *p* < 0.01, *η*^2^ = 0.18). There was no difference in other emotions between the two conditions.

#### 6.2.2. Small-Self Serves as a Mediator

We conducted a two-way ANOVA in which the emotion condition (awe = 1, neutral = 0) and the small-self condition (dampened = 1, not dampened = 0) were the independent variables, the negative emotion was the dependent variable, and demographics were the control variables. The interaction effect of the emotion condition and small-self condition was significant (*F* (1, 245) = 5.59, *p* = 0.02, *η*^2^ = 0.02). As shown in Figure 4, in the small-self not-dampened condition, the participants reported lower negative emotions in the awe condition (*M* = 2.36, *SD* = 1.55) versus the neutral condition (*M* = 3.61, *SD* = 1.84; *F* (1, 245) = 16.08, *p* < 0.01, *η*^2^ = 0.07). Nevertheless, in the small-self dampened condition, there was no significant difference in negative emotions arising from self-threat between the awe condition (*M* = 3.13, *SD* = 1.73) and the neutral condition (*M* = 3.30, *SD* = 1.81; *F* (1, 245) = 0.42, *p* = 0.52, *η*^2^ = 0.001).

Taken together, the awe played a buffering role on negative emotions arising from self-threat when the small-self was not dampened, whereas the buffering effect of awe was attenuated when the small-self was dampened. These findings ascertained that small-self drove the link between the awe and reduced negative emotions arising from self-threat. Thus, H2 was repeatedly verified.

## 7. General Discussion

Negative emotions arising from self-threat shape mental and physical health [1,5,6]. The issue of how to cope with this problem is attracting increasing academic attention. Our research provides empirical evidence of the buffering effect of awe on negative emotions arising from self-threat and its mediation mechanism. Across four studies, we verified that participants in the awe condition exhibited lower levels of negative emotions arising from self-threat, whether compared with those in the neutral (Study 1) or happiness conditions (Study 2). In addition, Study 1 also showed that the trait of awe did not affect the state of awe elicited by a video and negative emotions arising from self-threat. Finally, lending support to the second hypothesis, stimulation of the small-self explains the reason why awe alleviates negative emotions arising from self-threat (Study 3), even after controlling for the trait of awe and baseline moods (follow-up study). Thus, if the small-self is dampened, the buffering effect of awe on negative emotions is inhibited (Study 4). We strengthened the robustness of our findings by using different manipulations of awe, various self-threatening situations and manipulations of self-threat, distinct operationalization of the small-self, and involving participants from both individualistic and collectivistic cultures.

### 7.1. Theoretical Implications

In terms of theoretical implications, our research contributes in four aspects. First, our work complements earlier research on strategies for coping with negative emotions arising from self-threat. Previous research has mainly focused on direct resolution or fluid compensation [12,49], but these strategies are not available in some contexts or come with some negative side effects [2,11,17,18]. To address this void, we, from the perspective of distraction and reconstructing cognition, identify an original and malleable buffer—experiencing awe, which motivates people to change appraisals of the self and reduce the focus on the self and thus reduce the levels of negative emotions arising from self-threat. It is valuable for future work to examine whether awe similarly alleviates negative emotions towards major life issues like death or bankruptcy and explore why it might or might not work.

Second, our research contributes to the research on awe by extending its impacts on mental health. Prior research has documented that awe stimulates some cognitions such as perceived uncertainty, self-diminishment, and self-transcendence [21,22,32]. Adding to this stream of research, we verify the effect of awe on boosting the sense of small-self. Additionally, awe has been indicated to be effective in increasing some behaviors, such as prosocial behaviors [22] and global citizenship identification [23], whereas little is known about how awe impacts one’s mental and physical health. Our work addresses this gap by pivoting to uncover that awe, beyond affecting cognitive processes and behaviors, plays a significant role in alleviating negative emotions arising from self-threat that shapes both mental and physical health [1,5,6].

Third, our work is devoted to knowledge of the small-self. Previous studies have found that self-distracting [50], self-distancing [51], and mindfulness [35] could relieve distress or other negative reactions. Meanwhile, some researchers also supported the idea that the small-self is related to those above states [22,35], whereas they paid little attention to testing whether the small-self can make contributions to relieving negative reactions and how to activate the small-self’s buffering effect through external stimuli. Our research enriches the existing work by examining that awe evokes the small-self and thus alleviates the negative emotions arising from self-threat.

Fourth, the current research advances the testing of different states within a wide range of positive emotions. Beyond focusing on distinguishing negative emotions like anxiety, depression, and disgust [1], studying discrete positive emotions is attracting increasing academic attention [52]. Our findings support existing studies about positive emotions reducing some negative reactions [52,53]. Meanwhile, by comparing awe with other positive emotions, our research indicates that awe has unique impacts on negative emotions arising from self-threat through stimulating a sense of small-self, which inspires future research to explore other discrete positive emotions (e.g., gratefulness) and underlying mechanisms through which they alleviate negative reactions.

### 7.2. Practical Implications

In practical terms, this research contributes in three aspects. First, our findings inspire people to mitigate the interference of negative emotions by immersing themselves in awe conditions when encountering self-threat, thus promoting well-being [30,43]. Second, our research suggests that amid alleviating negative emotions arising from self-threat by relying on the cues of awe, individuals should attentively perceive and embrace the sense of the small-self, thus maximizing the driving role of the small-self. Third, the additional findings that the trait of awe did not interfere with the state of awe elicited by video and the alleviating effect of the state of awe [54] demonstrates that people with a low trait of awe can still try to evoke awe within a short period and thus achieve the relief of negative emotions arising from self-threat.

### 7.3. Limitations and Future Directions

Although our research makes some contributions, several limitations exist and indicate prospective avenues for future studies. First, since our method of manipulating emotion and self-threat is singular, more methods of inducing awe can be added, including field scenes, daily diaries, or other unnatural manipulated materials (e.g., great person and deep knowledge) [22,29,30], as well as more means of manipulating self-threats, such as storytelling and journaling [11,13]. Second, we verify that awe is more effective than happiness but do not explore why. Future research could explore whether the reason is the level of the evoked small-self. Meanwhile, the buffering effect of awe deserves to be further examined by comparing awe with more positive emotions such as optimistic, inspired, and relaxed emotions [21,31]. Finally, although our research regards awe as a positive emotion, negatively valanced awe elicited by some threatening scenes like volcanic eruptions, hurricanes, or earthquakes is common [55]. Negative awe may lead to lower well-being than the typical positive awe due to stimulating a sense of powerlessness [56], thus causing results opposite to the current findings. Alternatively, it is also possible that negative awe is still likely to impact people’s self-appraisals and thereby mitigate the negative emotions arising from self-threat. Future research deserves to pit these guesses against one another.

## 8. Conclusions

People usually perceive awe when facing something that is imposing, vast, or sublime. A sense of self as small and insignificant ensues, which helps people shift their attention from the self to larger entities and spurs the feeling that self-deficit is insignificant. Such a cognitive tendency makes people ignore or accept their self-deficit, thus alleviating negative emotions in self-threatening situations.

## Figures and Tables

**Figure 1 behavsci-13-00044-f001:**
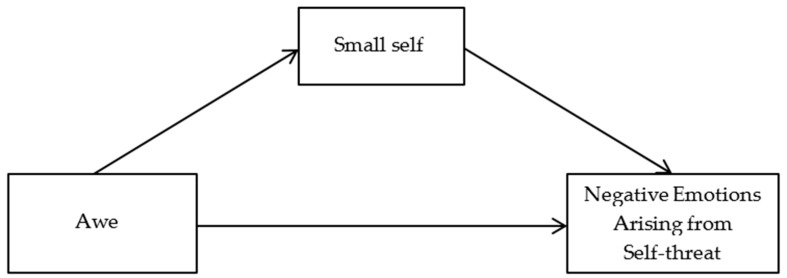
Conceptual model.

**Figure 2 behavsci-13-00044-f002:**
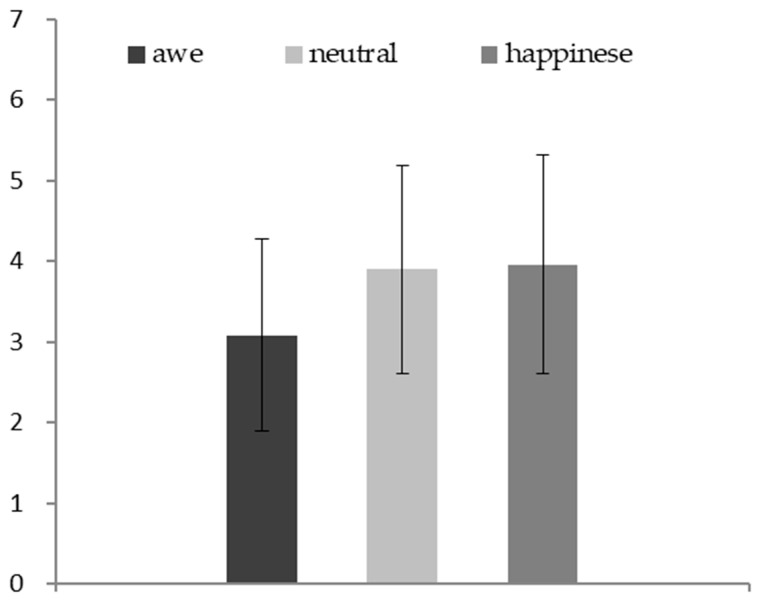
The effect of awe (vs. neutral vs. happiness) on negative emotions arising from self-threat.

**Figure 3 behavsci-13-00044-f003:**
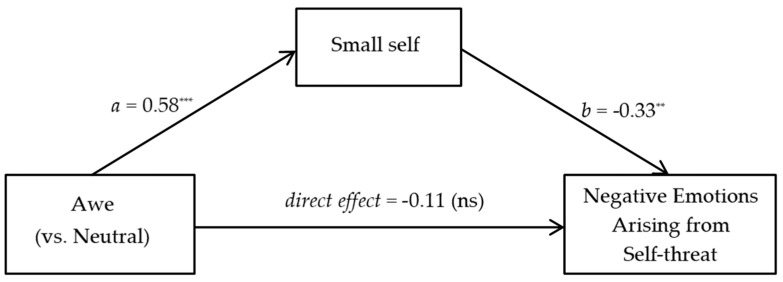
Results of mediation analysis. Notes: ** *p* < 0.01, *** *p* < 0.001, ns = not significant.

**Figure 4 behavsci-13-00044-f004:**
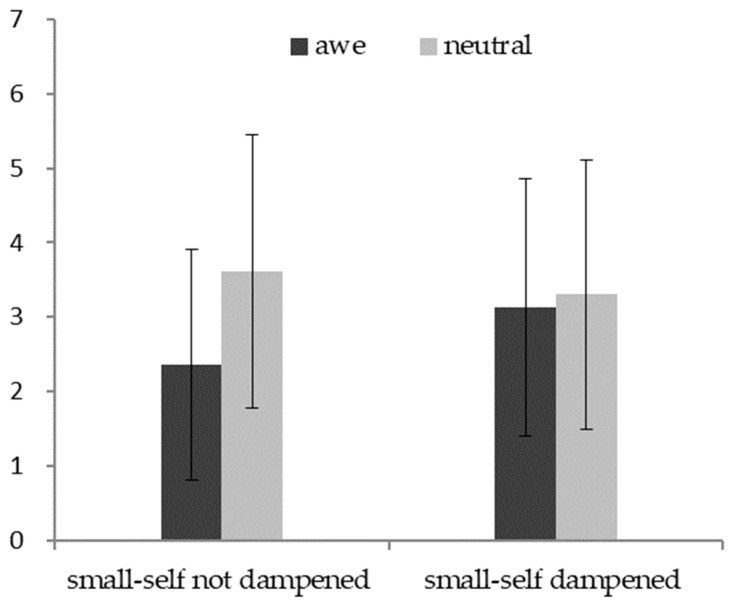
The interaction effect of awe (vs. neutral) and small-self (not dampened vs. dampened) on negative emotions arising from self-threat.

**Table 1 behavsci-13-00044-t001:** Overview of the four studies.

Study	Participants	Design	Self-Threatening Situation	Emotion Induction	Findings
Study 1	151 Chinese citizens	2 (emotion induction: awe vs. neutral)	Intelligence-threatening situation	Video-based induction	Support H1 Rule out the possibility that trait awe interferes with the degree to which state awe is elicited by video
Study 2	180 MTurk participants	3 (emotion induction: awe vs. neutral vs. happiness)	Appearance-threatening situation	Video-based induction	Support H1
Study 3	147 Chinese undergraduates	2 (emotion induction: awe vs. neutral)	Appearance-threatening situation	Situation-recalling induction	Support H2
Study 4	252 MTurk participants	2 (emotion induction: awe vs. neutral) * 2 (small-self: dampened vs. not dampened)	Power-threatening situation	Situation- recalling induction	Support H2

## Data Availability

The data presented in this study are available upon request from the corresponding author.

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
