# Peer review of "The Buffering Effect of Awe on Negative Emotions in Self-Threatening Situations"

_behavsci, 2023, doi:10.3390/bs13010044_

Round 1

Reviewer 1 Report

The introduction is informative and well written. Indeed the authors are able to address the hypotheses and guide the reader through a very complex psychological set of processes. Indeed I agree with the authors about the relationship between the awe and the pandemic. Despite the methodology, that I found very accurate, I advise de authors to add a specific figure or a flowchart depicting the whole process and the 4 studies. According to me, this is crucial in order to remember all the stages of the study and, overall, it should be helpful to replicate the study or part of it. Moreover, I agree with the authors about the 2 figures. I advise to add the SD bars to the histograms. You can choose to add std. err. or SD. Please, explain if the groups are completely independent among the 4 studies.

Despite the well written introduction, I found the Discussion quite short in terms of quality. I advise the authors to add more information to the discussion. Please, discuss study by study and then add a general discussion. In the current form, the discussion is difficult to follow .

Reviewer 2 Report

1. The excerpt from the introduction given below is unnecessary. I suggest removing it. 

(verses 31-35) "Especially, in the era of developed social media, it is easier for people to know about others’ achievements and perceive their shortcomings through WeChat or Facebook, in which users usually post something with the purpose of showing off [3,4]. Moreover, the COVID-19 epidemic has created so much damage to society that it has placed people in a situation of greater competition and frustration. Therefore, individuals experience more frequent and severe self-threat nowadays".

2. I am bothered by the lack of tables describing the statistical details of each study.

Round 2

Reviewer 1 Report

Please add the code and the institutional ethics committee that approved your study. Moreover, you are invited to add a version of of the informed consent.